# A Comparison between Vitamin D_3_ and 25-Hydroxyvitamin D_3_ on Laying Performance, Eggshell Quality and Ultrastructure, and Plasma Calcium Levels in Late Period Laying Hens

**DOI:** 10.3390/ani12202824

**Published:** 2022-10-18

**Authors:** Xiaoqing Jing, Yiwei Wang, Fulong Song, Xianfeng Xu, Mingkang Liu, Yu Wei, Huiling Zhu, Yulan Liu, Jintao Wei, Xiao Xu

**Affiliations:** 1Hubei Key Laboratory of Animal Nutrition and Feed Science, Wuhan Polytechnic University, Wuhan 430023, China; 2Wuhan Hualuo Branch, China Animal Husbandry Industry Co., Ltd., Wuhan 430000, China; 3Key Laboratory of Animal Embryo Engineering and Molecular Breeding of Hubei Province, Wuhan 430023, China

**Keywords:** vitamin D_3_, 25-hydroxyvitamin D_3_, performance, egg quality, laying hens

## Abstract

**Simple Summary:**

The sharp decline of laying performance and eggshell quality is a common problem in late period laying hens. Vitamin D_3_ (VD_3_) is a necessary micronutrient which plays an important role in mineral and skeletal homeostasis. With the rapid development of genetic selection, commercial laying hens have increased requirements for performance and nutrients. The commercial supplementary dose of VD_3_ (62.5 µg/kg) for late period laying hens may be not enough to satisfy the production. In addition, 25-hydroxyvitamin D_3_ (25-OHD_3_) as an active metabolite of VD_3_, is a viable alternative to replace VD_3_. Therefore, the objective of this study was to compare the effects of different supplementary doses and sources between VD_3_ and 25-OHD_3_ on the laying performance and eggshell quality in late period laying hens. The results showed that supplementary 125 µg/kg doses of VD_3_ or 25-OHD_3_ had better effects in late period laying hens compared with 62.5 µg/kg doses of VD_3_. Additionally, there were no different effects on laying performance or eggshell quality in the hens fed dietary 125 µg/kg doses of VD_3_ or 25-OHD_3_.

**Abstract:**

The objective of this study was to compare high supplementary doses (125 µg/kg) of vitamin D_3_ (VD_3_) or 25-hydroxyvitamin D_3_ (25-OHD_3_) with commercial supplementary doses (62.5 µg/kg) of VD_3_ on laying performance, eggshell quality and ultrastructure, and plasma calcium levels in late period laying hens. A total of 1512 Roman Gray (60-week-old) laying hens were allotted into three treatments with 12 replicates and 42 birds in each replicate. During the 12-week trial period, the layers were fed a basal diet supplemented with different doses of VD_3_ or 25-OHD_3_ (62.5 µg/kg VD_3_ in control group, CON; 125 µg/kg VD_3_ in high level VD_3_ group, VD_3_; 125 µg/kg 25-OHD_3_ in high level 25-OHD_3_ group, 25-OHD_3_). The results showed that high supplementary doses of VD_3_ or 25-OHD_3_ increased laying rate (*p* < 0.05). Moreover, the layers fed high doses of VD_3_ or 25-OHD_3_ diets had decreased unqualified egg rate and mortality (*p* < 0.05). High supplementary doses of VD_3_ or 25-OHD_3_ increased eggshell strength and eggshell thickness (*p* < 0.05). From observation in eggshell ultrastructure, high doses of VD_3_ or 25-OHD_3_ diets increased the palisade layer thickness and mammillary knob density (*p* < 0.05). Furthermore, high doses of VD_3_ or 25-OHD_3_ diets increased the calcium levels in plasma (*p* < 0.05). In summary, compared with 62.5 µg/kg doses of VD_3_, supplementary 125 µg/kg doses of VD_3_ or 25-OHD_3_ improved the laying performance, eggshell quality, and plasma calcium levels in late period laying hens. Additionally, there was an equal effect on laying performance and eggshell quality in the hens fed dietary 125 µg/kg doses of VD_3_ or 25-OHD_3_.

## 1. Introduction

Nowadays, the age of commercial laying hens can be extended beyond 80 weeks. However, a sharp decline of laying performance often occurs during the late period of laying hens [1]. This decrease in laying performance mainly includes a decreased laying rate and egg quality, such as eggshell quality and Haugh unit, and an increased feed–egg ratio [2,3]. Furthermore, during late laying period the hens often show calcium/phosphorus and lipid metabolism disorders which further result in negative effects on the health and laying performance of hens [4,5].

Generally, the negative effects on performance and egg quality of hens in the late laying period are mitigated by nutritional strategy [6]. Vitamin D_3_ (VD_3_) is a fat-soluble vitamin, and it plays an important role in mineral and skeletal homeostasis [7,8]. Previous studies and the NRC (1994) showed that the recommended minimum requirement of VD_3_ for laying hens was 300 IU/kg (equal to 7.5 µg/kg) [9,10]. In the current commercial production, the diets for laying hens often contains a higher VD_3_ level of 2500 IU/kg (62.5 µg/kg). Additionally, some studies showed that high dietary levels of VD_3_ improved eggshell quality and bone development [11,12]. However, with the rapid development of genetic selection, commercial laying hens will have improved production performance, leading to higher nutrient requirements. Currently, it is not clear whether increasing dietary VD_3_ level could extend the peak laying period and alleviate eggshell quality reduction during the late period of laying hens.

25-hydroxyvitamin D_3_ (25-OHD_3_), also known as 25-hydroxycholecalciferol, is an active metabolite of VD_3_ [13]. 25-OHD_3_ has been approved as a source of vitamin D in the poultry industry since 2006 and is widely applied nowadays [14]. Several studies showed that 25-OHD_3_ significantly improved laying performance, as well as the egg quality of laying hens [15,16]. However, other studies demonstrated that 25-OHD_3_ had no beneficial effect on laying performance and egg quality in laying hens [17,18]. The inconsistency of these results might be due to inclusion period of 25-OHD_3_, the dose supplemented, the breeds evaluated, and the laying stage. Moreover, previous studies showed that the recommended supplementary doses of 25-OHD_3_ were 35–225 µg/kg [14,19,20].

Based on the above studies, we hypothesized that dietary supplementation with high levels of VD_3_ or 25-OHD_3_ during the late laying period could improve the laying performance and egg quality of laying hens in the current commercial poultry industry. Therefore, the objective of this study was to compare the effects of different supplementary doses and sources, between VD_3_ and 25-OHD_3,_ on the laying performance and eggshell quality in late period laying hens.

## 2. Materials and Methods

### 2.1. Birds and Experimental Design

The experimental protocol (No. WPU202218028) used in this study was approved by the Institutional Animal Care and Use Committee of Wuhan Polytechnic University (Wuhan, China). A total of 1512 healthy hens (Roman Gray, 60-week-old) with similar body weight and egg-laying rates were used. This trial lasted for 12 weeks. All the birds were allotted to 3 dietary treatments, and each treatment included 12 replicates with 42 birds per replicate distributed in a completely randomized design. The treatments included a basal diet supplemented with VD_3_ at a commercial level (CON, 62.5 µg/kg diet), a basal diet supplemented with a high level of VD_3_ (VD_3_, 125 µg/kg diet), and a basal diet supplemented with a high level of 25-OHD_3_ (25-OHD_3_, 125 µg/kg diet). The basal diet was formulated in accordance with the nutrient requirements of the NRC (1994) and modified by Chinese standards (NY-T 33-2004) [10,21]. The ingredient composition and nutrient levels of the basal diet are shown in Table 1. Layers were raised at Hubei Xinhe Farm Co. LTD (Jingzhou, China). In the same layer house, the birds were fed in a 3-tiered three-dimensional cage (each replication consisted of 14 cages of 3 layers each, with each cage measuring 45 cm × 30 cm × 30 cm), with feed and water provided ad libitum, with a temperature range of 22–26 °C, a relative humidity range of 50–70%, and illumination for 16 h/D. The layers were standardized by initial body weight and egg production, and were set a 2-week adjustment to the experimental diets before starting the trial.

### 2.2. Laying Performance and Eggshell Quality

During the whole 12-week experimental period, egg number, total egg weight, unqualified eggs (eggs with a soft shell, broken shell, or different shape) were recorded daily for each replicate. At the end of 4, 8, and 12 weeks, the feed bags were weighed for each replicate to calculate the average daily feed intake (ADFI). Then, the laying rate, feed–egg ratio, and unqualified egg rate were calculated. The health status of the birds was recorded daily for each replicate to calculate mortality.

At the end of 4, 8, and 12 weeks, 72 normal eggs were randomly collected (24 eggs from each treatment, 2 eggs per replicate) to determine average egg weight and eggshell quality traits. Eggshell quality measurements included eggshell strength, eggshell thickness, and the percentage of organic matter, inorganic matter, calcium, and phosphorus in eggshell. Eggshell strength was determined by an EA-01 egg analyzer (ORKA Food Technology, Ltd., Ramat Hasharon, Israel) referring to the instructions of the manufacturer. Eggshell thickness was measured using a micrometer (SanLiang Precision instrument Co. Ltd., Guangdong, China) from 3 points, namely the blunt end, equatorial region, and sharp end [22]. The percentage of organic matter, inorganic matter, calcium, and phosphorus in eggshell was determined according to the procedures of the Association of Official Analytical Chemists (AOAC) [23].

### 2.3. Eggshell Ultrastructure

At the end of 4, 8, and 12 weeks, one egg per replicate was collected to evaluate eggshell ultrastructure. First, the eggs were broken, and the eggshell was cleaned with distilled water and dried for 48 h at room temperature. Then, two eggshell samples of each egg (0.5–1.0 cm^2^) were used for scanning electron microscope analysis (model TM 1000, HITACHI Corp., Chiyoda, Tokyo, Japan). One sample was used for the analysis of the cross section of the eggshell, and the other sample was used for the measurement of the internal surface. Before scanning, the eggshell samples were immobilized to the aluminum support and sprayed with gold powder, as done in the previous study [24].

For the cross-sectional measurement, scanned images were obtained for each sample using 200× magnification. The thickness of mammillary layer, palisade layer, crystal layer and cuticle, and the width of the mammillary knobs in the cross section, were measured following the procedure of the previous study [25]. The number of mammillary knobs distributed within 1 mm^2^ area of the shell surface was counted to calculate the density of mammillary knobs [26].

### 2.4. Plasma Calcium and Phosphorus Levels

At the end of 4, 8, and 12 weeks, one layer per replicate was randomly selected to collect blood samples. Blood was collected via sub wing vein (approximately 5 mL). Then, the plasma samples were extracted and stored at −80 °C after centrifugation at 3500 rmp for 10 min. Plasma calcium and phosphorus levels were determined by an automatic biochemical analyzer (7100, HITACHI, Tokyo, Japan) according to the methods of the previous study [27].

### 2.5. Statistical Analyses

All data were analyzed as a randomized block design using the general linear model procedures (GLM) of SAS (SAS Institute Inc., Cary, NC, USA). The replication was the experimental unit for all parameters. The results were considered significant at *p* < 0.05. If significant effects were found, individual means were compared using Duncan’s multiple comparison tests. Results are expressed as least squares means and standard error of the mean (SEM).

## 3. Results and Discussion

### 3.1. Laying Performance

The results of the laying performance are shown in Table 2. During weeks 1–4, the hens fed VD_3_ and 25-OHD_3_ diets had significantly decreased unqualified egg rates and mortality compared with the hens fed the control diets (*p* < 0.05). However, the laying rate and feed–egg ratio had no significant difference among the three treatments (*p* > 0.05). During weeks 5–8 and 9–12, the hens fed VD_3_ and 25-OHD_3_ diets had significantly improved laying rates, and decreased unqualified egg rates and mortality compared with the hens fed the control diets (*p* < 0.05). During the whole experimental period, dietary supplementation with VD_3_ and 25-OHD_3_ significantly enhanced laying rates, and reduced unqualified egg rates and mortality of the hens compared with the control group (*p* < 0.05). Optimum levels of vitamins in poultry diets allow the birds to perform according to their genetic potential. The requirements of VD_3_ for layers were established decades ago, so the recommended requirement data do not satisfy the current genetically superior birds with increased growth, laying performance, and feed efficiency [14,18]. VD_3_ and 25-OHD_3_ are widely used as vitamin D sources in the poultry industry. However, the proper supplementary doses of the two sources are not consistent in the previous reports. Some studies showed that no difference in egg production improvement was found between late period laying hens fed VD_3_ or 25-OHD_3_ diets [14,28]. Conversely, some studies showed dietary 25-OHD_3_ (75 µg/kg) improved laying performance in laying hens [29]. In our study, we found that high dietary doses of VD_3_ and 25-OHD_3_ (125 µg/kg) both improved layers’ performance compared to commercial VD_3_ doses (62.5 µg/kg). The current results demonstrate that high levels of VD_3_ or 25-OHD_3_ should be used in the practical poultry industry to meet the layers’ requirements. The reason for similar performance between the hens fed VD_3_ and 25-OHD_3_ diets may be that the supplementary doses of the two sources both satisfy the layers’ requirement for vitamin D. Different studies showed that the layers aged 60 weeks had a large variation in laying performance [2,6]. In future research, a longer trial period, which reaches the sharp decline of laying rate, should be adopted. The effect of VD_3_ and 25-OHD_3_ on laying performance may be reflected more significantly in the older layers.

### 3.2. Eggshell Quality

Table 3 shows the effects of vitamin D_3_ and 25-hydroxyvitamin D_3_ on eggshell quality in late period laying hens. At the end of the 4th week, the hens fed VD_3_ diets had significantly increased eggshell thickness compared with the hens fed the control diets (*p* < 0.05). At the end of the 8th week, the hens fed 25-OHD_3_ diets had significantly increased eggshell strength compared with the hens fed the control diets (*p* < 0.05). Moreover, the hens fed VD_3_ and 25-OHD_3_ diets had significantly increased eggshell thickness compared with the hens fed the control diets (*p* < 0.05). At the end of the 12th week, dietary supplementation with VD_3_ and 25-OHD_3_ both significantly increased eggshell strength and eggshell thickness compared with the control group (*p* < 0.05). About 10–15% of eggs are lost as a result of eggshell quality problems before or during egg collection, leading to severe economic losses [30]. Therefore, eggshell quality is a major concern in the commercial egg industry. With the increase of laying age, the eggshell strength and thickness are often reduced [31]. Previous studies demonstrated that increasing dietary VD_3_ could enhance the eggshell quality [32]. In agreement with the previous studies, our experiment showed the positive effect on eggshell strength and thickness of hens fed high levels of VD_3_ or 25-OHD_3_. VD_3_ could improve eggshell quality through regulating calcium/phosphorus metabolism and enhancing endometrium morphology [26]. However, there were few reports showing the effect of 25-OHD_3_ on the eggshell quality. Our results demonstrated high levels of VD_3_ and 25-OHD_3_ had a similar function on the eggshell quality in late period laying hens.

### 3.3. Eggshell Ultrastructure

The results of eggshell ultrastructure measured by scanning electron microscope are shown in Table 4. At the end of the 4th week, the hens fed 25-OHD_3_ diets had a significantly increased palisade layer thickness compared with the hens fed the control diets (*p* < 0.05). The hens fed VD_3_ and 25-OHD_3_ diets had significantly increased mammillary knob density compared with the hens fed the control diets (*p* < 0.05). At the end of the 8th week, the hens fed VD_3_ and 25-OHD_3_ diets had a significantly increased palisade layer thickness and mammillary knob density compared with the hens fed the control diets (*p* < 0.05). Moreover, the hens fed 25-OHD_3_ diets had a significantly decreased mammillary knob width compared with the hens fed the control diets (*p* < 0.05). At the end of the 12th week, the hens fed VD_3_ and 25-OHD_3_ diets had a significantly increased palisade layer thickness and mammillary knob density, as well as significantly decreased mammillary knob width compared with the hens fed the control diets (*p* < 0.05). Figure 1, Figure 2 and Figure 3 show the representative picture of the eggshell ultrastructure of the hens fed the three diets. These figures also show that the layers fed VD_3_ and 25-OHD_3_ diets had enhanced eggshell ultrastructure compared with the layers fed the control diets. The eggshell is formed in the uterus with a 10–12 h mineralization after ovulation. The stages of mineralization are in the order of mammillary layer formation, linear calcification, and termination [33]. Therefore, the eggshell is divided into mammillary layer, palisade layer, crystal layer, and cuticle from the inside out [34]. Some studies reported that the total thickness of the eggshell, palisade layer thickness, and the density and width of mammillary knobs affected the strength of the eggshell [35]. The palisade layer consists of calcium carbonate calcite crystals, accounting for two-thirds of the total thickness of the eggshell [36]. Our results showed that high levels of VD_3_ and 25-OHD_3_ increased the thickness of palisade layer and density of mammillary knobs which is consistent with previous studies. These reported that high-quality eggshells tended to have a larger thickness of palisade layer and density of mammillary knobs [19]. The results are also in agreement with the improvement of the whole eggshell thickness of the hens fed diets with high levels of VD_3_ and 25-OHD_3_. Therefore, the evaluation of the eggshell ultrastructure provides us a good understanding of the eggshell structure and quality.

### 3.4. Plasma Calcium and Phosphorus Levels

Table 5 shows the effects of vitamin D_3_ and 25-hydroxyvitamin D_3_ on plasma calcium and phosphorus levels in late period laying hens. The hens fed VD_3_ and 25-OHD_3_ diets all had significantly increased plasma calcium levels compared with the hens fed the control diets at the end of 4th, 8th, and 12th weeks (*p* < 0.05). Moreover, the hens fed the 25-OHD_3_ diet had significantly increased plasma phosphorus levels compared with the hens fed the VD_3_ diet at the end of 4th week (*p* < 0.05). However, there was no significant difference in the plasma phosphorus levels among the three groups at the end of 8th or 12th week (*p* > 0.05). VD_3_ plays an important role in the proper metabolism of calcium and phosphorus. Previous studies showed that VD_3_ enhanced calcium and phosphorus absorption and metabolism in the intestine and bones [14,37]. For laying hens, calcium is one of the key nutrients required for optimal eggshell quality for shell formation and calcification [38]. Low egg production and eggshell quality are associated with low calcium and phosphorus utilization and VD_3_ deficiency in laying hens [39]. Our study showed that high levels of VD_3_ and 25-OHD_3_ increased plasma calcium levels in late period laying hens, which demonstrates that the hens in the VD_3_ and 25-OHD_3_ groups had better calcium utilization. The increased plasma calcium levels are in agreement with the enhanced laying performance and eggshell quality in the current study. Based on the results of this study, the recommended dose of VD_3_ in the diet can be increased for late period laying hens. In future research, a smaller supplementary dose of 25-OHD_3_ replacing a high level of VD_3_ in diets should be studied, to clarify the inconsistent views on the higher bioavailability of 25-OHD_3_ to VD_3_ [15,18,40].

## 4. Conclusions

In conclusion, compared with 62.5 µg/kg doses of VD_3_, supplementary 125 µg/kg doses of VD_3_ or 25-OHD_3_ improved the laying performance, eggshell quality, and plasma calcium levels in late period laying hens. Moreover, there was an equal effect on the laying performance and eggshell quality in the hens fed dietary 125 µg/kg doses of VD_3_ or 25-OHD_3_.

## Figures and Tables

**Figure 1 animals-12-02824-f001:**
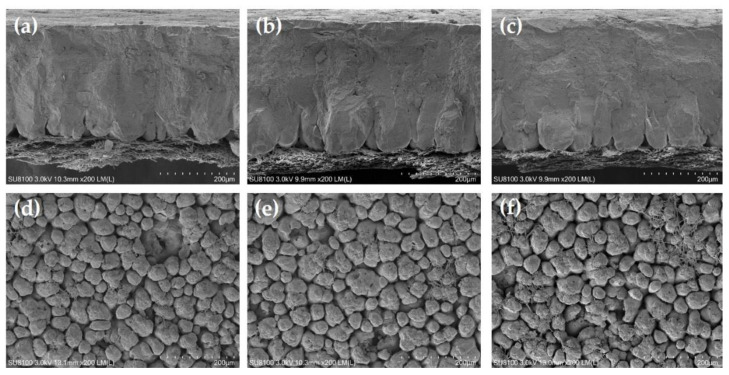
Scanning electron microscopy of the eggshell cross section and internal surface of the laying hens at 4 weeks of age. (**a**–**c**) show the cross section with CON, VD_3_, and 25-OHD_3_, respectively. (**d**–**f**) show the internal surface with CON, VD_3_, and 25-OHD_3_, respectively. Scale bar: 200 μm.

**Figure 2 animals-12-02824-f002:**
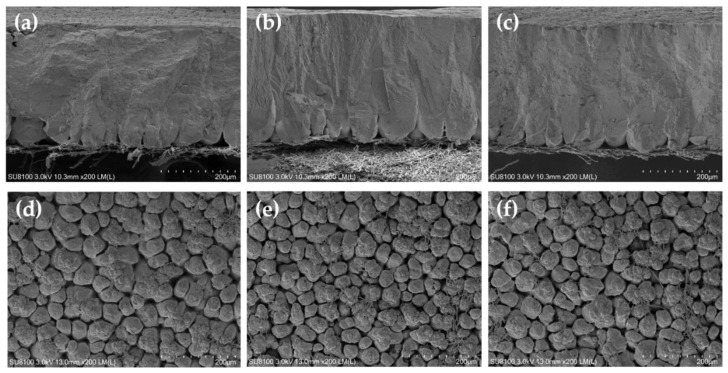
Scanning electron microscopy of the eggshell cross section and internal surface of the laying hens at 8 weeks of age. (**a**–**c**) show the cross section with CON, VD_3_, and 25-OHD_3_, respectively. (**d**–**f**) show the internal surface with CON, VD_3_, and 25-OHD_3_, respectively. Scale bar: 200 μm.

**Figure 3 animals-12-02824-f003:**
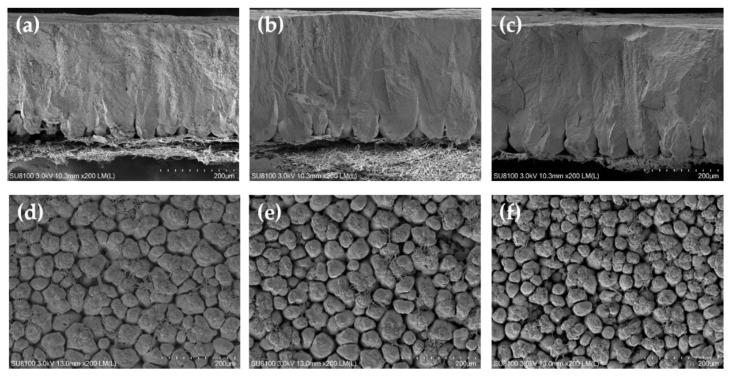
Scanning electron microscopy of the eggshell cross section and internal surface of the laying hens at 12 weeks of age. (**a**–**c**) show the cross section with CON, VD_3_, and 25-OHD_3_, respectively. (**d**–**f**) show the internal surface with CON, VD_3_, and 25-OHD_3_, respectively. Scale bar: 200 μm.

**Table 1 animals-12-02824-t001:** Composition and nutrient contents of basal diet for the hens during late laying period (as fed basis).

Ingredients, %		Nutrient Contents, % ^2^	
Corn	66.35	Metabolizable energy, MJ/kg	11.70
Soybean meal	22.50	Crude protein	15.68
Rice bran	0.60	Calcium	4.06
Limestone	8.00	Phosphorus	0.69
Zeolite	0.20	Lysine	0.68
Calcium hydrophosphate	0.90	Methionine	0.40
Salt	0.35		
DL-methionine	0.10		
Vitamin and mineral premix ^1^	1.00		
Total	100.00		

^1^ Each kg of the premix contains: vitamin A, 8500 IU; vitamin E, 17.5 IU; vitamin K_3_, 2.2 mg; thiamine, 2.5 mg; riboflavin, 7.8 mg; pyridoxine, 4.0 mg; cobalamin, 17.6 μg; nicotinic acid, 27.5 mg; pantothenic acid, 14.0 mg; folic acid, 0.8 mg; biotin, 90 μg; choline chloride, 350 mg; iron, 80 mg; zinc, 96.6 mg; manganese, 100 mg; copper, 13 mg; selenium, 0.3 mg; iodine, 2.65 mg; cobalt, 0.3 mg. ^2^ The nutrient contents are analyzed values except metabolizable energy which are calculated values.

**Table 2 animals-12-02824-t002:** Effects of vitamin D_3_ and 25-hydroxyvitamin D_3_ on laying performance in late period laying hens ^1^.

Item	CON	VD_3_	25-OHD_3_	SEM	*p*-Value
1–4 week					
Laying rate, %	91.82	93.54	93.20	1.17	0.189
Average daily feed intake, g	134	135	133	4	0.584
Feed-egg ratio	2.23	2.15	2.16	0.05	0.095
Unqualified egg rate, %	0.25 ^a^	0.13 ^b^	0.13 ^b^	0.02	<0.001
Mortality, %	0.57 ^a^	0.21 ^b^	0.17 ^b^	0.04	<0.001
5–8 week					
Laying rate, %	89.08 ^b^	92.54 ^a^	91.68 ^a^	1.00	0.009
Average daily feed intake, g	136	137	136	3	0.979
Feed-egg ratio	2.32	2.25	2.25	0.06	0.564
Unqualified egg rate, %	0.26 ^a^	0.20 ^b^	0.19 ^b^	0.02	0.021
Mortality, %	0.71 ^a^	0.24 ^b^	0.18 ^b^	0.05	<0.001
9–12 week					
Laying rate, %	84.28 ^b^	87.41 ^a^	87.68 ^a^	1.11	0.014
Average daily feed intake, g	117	119	116	3	0.421
Feed-egg ratio	2.12	2.05	2.02	0.04	0.067
Unqualified egg rate, %	0.56 ^a^	0.27 ^b^	0.24 ^b^	0.04	<0.001
Mortality, %	0.44 ^a^	0.00 ^b^	0.00 ^b^	0.03	<0.001
1–12 week					
Laying rate, %	88.39 ^b^	91.18 ^a^	91.03 ^a^	0.84	0.006
Average daily feed intake, g	129	130	128	3	0.835
Feed-egg ratio	2.23	2.15	2.14	0.04	0.095
Unqualified egg rate, %	0.36 ^a^	0.20 ^b^	0.18 ^b^	0.03	<0.001
Mortality, %	1.72 ^a^	0.45 ^b^	0.35 ^b^	0.13	<0.001

SEM, standard error of the mean; CON, 62.5 µg/kg vitamin D_3_ in the diet; VD_3_, 125 µg/kg vitamin D_3_ in the diet; 25-OHD_3_, 125 µg/kg 25-hydroxyvitamin D_3_ in the diet. ^a,b^ Within a row means followed by different letters are different at *p* < 0.05. ^1^ There were 12 replicates per treatment.

**Table 3 animals-12-02824-t003:** Effects of vitamin D_3_ and 25-hydroxyvitamin D_3_ on eggshell quality in late period laying hens ^1^.

Item	CON	VD_3_	25-OHD_3_	SEM	*p*-Value
4th week					
Egg weight, g	65.30	65.55	65.25	0.37	0.837
Eggshell strength, N	38.51	43.58	42.64	2.49	0.164
Eggshell thickness, μm	402 ^b^	437 ^a^	434 ^ab^	16	0.046
Organic matter, %	3.50	3.46	3.52	0.15	0.825
Inorganic matter, %	96.50	96.54	96.48	0.18	0.892
Calcium, %	38.1	39.5	39.0	2.0	0.793
Phosphorus, %	0.42	0.37	0.38	0.02	0.308
8th week					
Egg weight, g	65.58	65.50	65.34	0.36	0.705
Eggshell strength, N	37.94 ^b^	41.99 ^ab^	46.38 ^a^	2.34	0.008
Eggshell thickness, μm	419 ^b^	451 ^a^	456 ^a^	9	<0.001
Organic matter, %	3.53	3.49	3.44	0.14	0.930
Inorganic matter, %	96.47	96.51	96.56	0.15	0.925
Calcium, %	38.2	39.8	40.5	2.2	0.752
Phosphorus, %	0.40	0.38	0.38	0.02	0.426
12th week					
Egg weight, g	65.62	65.66	65.29	0.37	0.714
Eggshell strength, N	37.06 ^b^	40.17 ^a^	41.98 ^a^	1.53	0.003
Eggshell thickness, μm	383 ^b^	415 ^a^	421 ^a^	15	0.015
Organic matter, %	3.48	3.58	3.55	0.15	0.937
Inorganic matter, %	96.52	96.42	96.45	0.17	0.901
Calcium, %	40.3	39.8	40.8	2.3	0.750
Phosphorus, %	0.41	0.40	0.37	0.02	0.551

SEM, standard error of the mean; CON, 62.5 µg/kg vitamin D_3_ in the diet; VD_3_, 125 µg/kg vitamin D_3_ in the diet; 25-OHD_3_, 125 µg/kg 25-hydroxyvitamin D_3_ in the diet. ^a,b^ Within a row means followed by different letters are different at *p* < 0.05. ^1^ There were 12 replicates per treatment.

**Table 4 animals-12-02824-t004:** Effects of vitamin D_3_ and 25-hydroxyvitamin D_3_ on eggshell ultrastructure (μm) in late period laying hens ^1^.

Item	CON	VD_3_	25-OHD_3_	SEM	*p*-Value
4th week					
Mammillary layer thickness, μm	117	128	123	6	0.473
Palisade layer thickness, μm	237 ^b^	254 ^ab^	268 ^a^	11	0.009
Crystal layer thickness, μm	20.0	21.4	22.8	1.8	0.862
Cuticle thickness, μm	7.2	7.8	8.0	0.6	0.669
Mammillary knob width, μm	82.5	75.6	77.4	4.2	0.620
Mammillary knob density, 1 mm^2^	142 ^b^	175 ^a^	190 ^a^	8	<0.001
8th week					
Mammillary layer thickness, μm	116	129	128	6	0.314
Palisade layer thickness, μm	236 ^b^	266 ^a^	273 ^a^	10	0.012
Crystal layer thickness, μm	20.4	22.4	22.8	1.7	0.601
Cuticle thickness, μm	7.0	7.5	7.9	0.5	0.418
Mammillary knob width, μm	81.0 ^a^	71.8 ^ab^	70.4 ^b^	4.0	0.042
Mammillary knob density, 1 mm^2^	138 ^b^	162 ^a^	167 ^a^	7	0.004
12th week					
Mammillary layer thickness, μm	112	128	126	7	0.719
Palisade layer thickness, μm	232 ^b^	259 ^a^	277 ^a^	10	<0.001
Crystal layer thickness, μm	21.1	22.5	23.6	1.8	0.479
Cuticle thickness, μm	7.2	7.6	8.1	0.5	0.798
Mammillary knob width, μm	79.8 ^a^	70.5 ^b^	68.5 ^b^	4.1	0.028
Mammillary knob density, 1 mm^2^	142 ^b^	157 ^a^	165 ^a^	7	<0.001

SEM, standard error of the mean; CON, 62.5 µg/kg vitamin D_3_ in the diet; VD_3_, 125 µg/kg vitamin D_3_ in the diet; 25-OHD_3_, 125 µg/kg 25-hydroxyvitamin D_3_ in the diet. ^a,b^ Within a row means followed by different letters are different at *p* < 0.05. ^1^ There were 12 replicates per treatment.

**Table 5 animals-12-02824-t005:** Effects of vitamin D_3_ and 25-hydroxyvitamin D_3_ on plasma calcium and phosphorus levels (mmol/L) in late period laying hens ^1^.

Item	CON	VD_3_	25-OHD_3_	SEM	*p*-Value
4th week					
Calcium	3.14 ^b^	3.88 ^a^	3.93 ^a^	0.22	0.008
Phosphorus	2.31 ^ab^	2.25 ^b^	2.62 ^a^	0.18	0.024
8th week					
Calcium	3.24 ^b^	3.75 ^a^	3.98 ^a^	0.22	0.035
Phosphorus	2.33	2.36	2.57	0.17	0.328
12th week					
Calcium	3.10 ^b^	3.62 ^a^	3.71 ^a^	0.20	0.015
Phosphorus	2.21	2.42	2.65	0.19	0.082

SEM, standard error of the mean; CON, 62.5 µg/kg vitamin D_3_ in the diet; VD_3_, 125 µg/kg vitamin D_3_ in the diet; 25-OHD_3_, 125 µg/kg 25-hydroxyvitamin D_3_ in the diet. ^a,b^ Within a row means followed by different letters are different at *p* < 0.05. ^1^ There were 12 replicates per treatment.

## Data Availability

The study did not report any data.

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
