# Peer review of "A Comparison between Vitamin D_3_ and 25-Hydroxyvitamin D_3_ on Laying Performance, Eggshell Quality and Ultrastructure, and Plasma Calcium Levels in Late Period Laying Hens"

_animals, 2022, doi:10.3390/ani12202824_

Round 1

Reviewer 1 Report

Better to incorporate the temperature and relative humidity data taken during research trial.

Author Response

The point-by-point response to the reviewer 1's comments was attached by a file.

Reviewer 2 Report

I reviewed the article A comparison between vitamin D3 and 25-hydroxyvitamin D3 2 on laying performance, eggshell quality and ultrastructure, and 3 plasma calcium levels in the late laying period hens. The topic is very intersting and provides novel information on the egg production and egg quality under two different sources of vitamin D. 

The conclusion in the abstract and the conclusion should be the same. 

add the following reference and take proper guideline from it. 

Dietary vitamin D: growth, physiological and health consequences in broiler production. Animal Biotechnology doi: https://doi.org/10.1080/10495398.2021.2013861

I think the experimetnal period is very small. Is it possible to add some more weeks?

in my opinion, the text of the results section is very long. I think it should be summarized. 

Author Response

The point-by-point response to the reviewer 2's comments was attached by a file.

Reviewer 3 Report

Please see the document attached. 

Author Response

The point-by-point response to the reviewer 3's comments was attached by a file.

Round 2

Reviewer 3 Report

Some of the changes were addressed correctly. However, the manuscript still needs editing in the sentence structure, verb agreement, and use of past and present tense. This manuscript does not meet the adequate standards for publication in an international journal and it should not be published unless it undergoes a professional editing service. The reviewer has added more “corrections” and comments to include in the editing.

Line 39: In summary, compared with 62.5 µg/kg doses of VD3, supplementary 125 µg/kg doses of VD3 or 25-OHD3 improved the laying performance, eggshell quality, and plasma calcium levels in the late period laying hens.

Line 55, remove “is a kind of”

Line 57: Previous studies and the NRC (1994) showed that the recommended minimum requirement of VD3 for laying hens were 300 IU/kg (equally to 7.5 µg/kg) [9-10].

Line 59: the diets, not the feed. Remove exceeded.

Line 63: performance, leading to a higher nutrient requirement. Remove: The improved performance often needs higher requirement for nutrients.

Line 63: “Currently, it is not clear whether increasing dietary VD3…”

Line 68: Several studies showed…. (Remove the majority of studies)

Line 70: However, other studies (Remove some).

Line 72: The inconsistency of these results might be due to inclusion period of 25-OHD3, the dose supplemented, the breeds evaluated, and the laying stage. Moreover, previous studies showed that the recommended….

Line 80: Remove (one) “different” … “different supplementary doses and sources of VD3 and 25-OHD3 on the laying performance and eggshell quality…

Line 81: … in late period laying hens. (Please revise this throughout the entire manuscript).

89: distributed in a completely randomized design.

Line 93: of the NRC (1994)  and…

Line 97: Consisted

Line 120: the percentage instead of the content?

Line 161 and throughout the manuscript: fed the control diets.

Line 169: The requirements of VD3 for layers were established decades ago, so the recommend 169 requirement data are not satisfy the current genetically superior birds with increased 170 growth, laying performance and feed efficiency [19,29].

Please re-phrase. The document, mostly the results and discussion section need to undergo a professional English editing service. This is not a small mistake; the quality of writing, spelling, and transition within sentences is not sufficient for publication in a peer review journal. The same applies for line 173, 175, 180, 287… and more…

Line 175, also add the dose in that study.

Table 4. A brief explanation was provided on measuring the wrong way, and the measurements were changed. However, the reviewer would like to have a more in-depth explanation of what measurements were not done correctly? Is this something that also needs to be specified in the materials and methods to avoid confusion for the reader?

Line 287: Suggestion… “Based on the results of this study, the recommended dose of VD3 in the diet can be increased for late period laying hens”.

Line 289 is hard to understand, please re-write. The entire section needs to be revised for spelling, grammar and overall sentence structure.

Author Response

Thank you for your reviewing work. We have revised all the concerns raised by the reviewer. The point-by-point response letter was attached by a file below. Thank you again and hope for your reply.
